# Chemerin and Chemokine-like Receptor 1 Expression in Ovarian Cancer Associates with Proteins Involved in Estrogen Signaling

**DOI:** 10.3390/diagnostics13050944

**Published:** 2023-03-02

**Authors:** Florian Weber, Susanne Schueler-Toprak, Christa Buechler, Olaf Ortmann, Oliver Treeck

**Affiliations:** 1Institute for Pathology, University of Regensburg, 93053 Regensburg, Germany; 2Department of Gynecology and Obstetrics, University Medical Center Regensburg, 93053 Regensburg, Germany; 3Department of Internal Medicine I, University Medical Center Regensburg, 93053 Regensburg, Germany

**Keywords:** chemerin, chemokine-like receptor 1, estrogen-related receptors, ovarian cancer, overall survival, progression-free survival

## Abstract

Chemerin, a pleiotropic adipokine coded by the *RARRES2* gene, has been reported to affect the pathophysiology of various cancer entities. To further approach the role of this adipokine in ovarian cancer (OC), intratumoral protein levels of chemerin and its receptor chemokine-like receptor 1 (CMKLR1) were examined by immunohistochemistry analyzing tissue microarrays with tumor samples from 208 OC patients. Since chemerin has been reported to affect the female reproductive system, associations with proteins involved in steroid hormone signaling were analyzed. Additionally, correlations with ovarian cancer markers, cancer-related proteins, and survival of OC patients were examined. A positive correlation of chemerin and CMKLR1 protein levels in OC (Spearman’s rho = 0.6, *p* < 0.0001) was observed. Chemerin staining intensity was strongly associated with the expression of progesterone receptor (PR) (Spearman´s rho = 0.79, *p* < 0.0001). Both chemerin and CMKLR1 proteins positively correlated with estrogen receptor β (ERβ) and estrogen-related receptors. Neither chemerin nor the CMKLR1 protein level was associated with the survival of OC patients. At the mRNA level, in silico analysis revealed low *RARRES2* and high *CMKLR1* expression associated with longer overall survival. The results of our correlation analyses suggested the previously reported interaction of chemerin and estrogen signaling to be present in OC tissue. Further studies are needed to elucidate to which extent this interaction might affect OC development and progression.

## 1. Introduction

Ovarian cancer (OC) is the leading cause of death by a gynecological malignancy in the developed world [1]. Due to missing screening methods and the aggressive behavior of the disease, the majority are diagnosed in advanced stages [2]. OC has a five-year survival rate of only 10% when the most common serous type spreads rapidly throughout the peritoneal cavity. Overall, this disease has a poor prognosis, with a five-year survival rate of approximately 50%. If diagnosed in earlier stages when the cancer is still confined to the ovary, this survival rate could rise to about 90%, but today this occurs in only 20% of patients [2,3].

Increasing evidence suggests that ovarian cancer, like tumors of different origins, is affected by adipokine chemerin [4,5,6]. Chemerin (RARRES2) is a well-described adipokine [7]. It was initially identified as a chemoattractant protein for immune cells that binds to chemokine-like receptor 1 (CMKLR1) expressed by these cells. In the meantime, diverse functions of chemerin have been defined, and chemerin was shown to regulate angiogenesis, adipogenesis, insulin response, and blood pressure [8,9,10,11,12,13]. Although with CCRL2 and GPR1, two further chemerin receptors have been identified, CMKLR1 has been considered to be the most important receptor of this adipokine since chemerin binding to CMKLR1 particularly leads to broad G-protein activation [14]. CMKLR1, located in the cell membrane, is internalized upon chemerin binding. Ligand binding initiates activation of G-proteins and β-arrestin pathways, inducing cellular responses via second messenger pathways such as intracellular calcium mobilization, phosphorylation of mitogen-activated protein kinase (MAPK)1/MAPK2 (ERK1/2), tyrosine-protein kinase receptor (TYRO) 3, MAPK14/p38 MAPK and phosphoinositid-3-kinase (PI3K) [14,15]. Emerging studies have proven the role of chemerin in tumorigenesis, whose expression often differs between tumor and non-tumor tissues [4,16]. In most tumor entities, chemerin/RARRES2 is down-regulated compared to normal tissue, e.g., in tumors of the breast, melanoma, lung, prostate, liver, adrenal, and in melanoma, and this decrease of chemerin expression has been suggested to be part of the tumor´s immune escape [4,17].

Estrogens are known to affect the progression of ovarian cancer [18], although to a much lesser extent than breast cancer. These effects are dependent on the expression of estrogen receptors (ERs) α and β. Estrogens activate the proliferation of ovarian cancer cells via ERα, often being overexpressed in this cancer entity [18,19]. Expression of ERβ, which is the predominant ER in the ovary [20], is often down-regulated in OC. ERβ is associated with an improved overall survival (OS) [21,22] in line with in vitro data demonstrating that its activation reduces ovarian cancer cell proliferation and activates apoptosis [21,23,24,25]. There is a relationship between estrogen-related receptors (ERRs) α, β, and γ with various cancer-related genes as well as ERα in ovarian cancer [26]. ERRs interact with ERα and several other nuclear receptors [27,28]. Thereby, among others, a vast number of different genes modulating metabolic processes are regulated, and several different pathways are controlled [29]. ERRα, which has attracted the greatest attention to date, acts as a master regulator of cellular metabolism, thereby also promoting tumor growth [30]. Chemerin was shown to decrease ovarian steroidogenesis via CMKLR1 [31,32] and thus may be protective in hormone-dependent cancers. A tumor-suppressive effect of chemerin was also reported by a recent in vitro study demonstrating chemerin to reduce the growth of ovarian cancer cell spheroids via activating the release of interferon (IFN)α, leading to induction of a broad, IRF9/ISGF3-mediated anti-tumoral transcriptome response [6]. However, a recent Chinese in vitro study reported a tumor-promoting role of chemerin in ovarian cancer cell lines in terms of proliferation via upregulation of programmed death ligand 1 (PD-L1) [5]. 

On the mRNA level, data on the expression of *RARRES2* and *CMKLR1* in ovarian cancer tissue have been extensively collected, e.g., by The Cancer Genome Atlas (TCGA) project https://www.cancer.gov/tcga). However, studies based on protein data of both genes in OC are rare. Thus, to further approach the possible role of chemerin and CMKLR1 in this cancer entity, analyses of their protein levels in OC cancer tissue and identification of correlated proteins are necessary. In the current study, protein levels of chemerin and CMKLR1 were assessed by immunohistochemistry of tissue microarrays (TMA), including tissues of 208 ovarian cancer patients. Furthermore, their association with patients´ survival and with the expression of ovarian cancer markers, cancer-related proteins, and components of estrogen signaling pathways was tested.

## 2. Materials and Methods

### 2.1. Tissue Samples

In this study, ovarian cancer samples collected in the Department of Pathology of the University of Regensburg were examined. Generally, Caucasian women with sporadic ovarian cancer and available information on grading, stage, and histological subtype from 1995 to 2013 were included. Patients’ clinical data were available from tumor registry database information provided by the Tumor Center Regensburg (Bavaria, Germany). This high-quality population-based regional cancer registry was founded in 1991, and it covers a population of more than 2.2 million people in Upper Palatinate and Lower Bavaria. Information about the diagnosis, course of the disease, therapies, and long-term follow-up are documented. Patient data originate from the University Hospital Regensburg, 53 regional hospitals, and more than 1000 practicing doctors in the region. Based on medical reports, pathology, and follow up-records, these population-based data are routinely documented and fed into the cancer registry (Table 1).

### 2.2. Tissue Microarray and Immunohistochemistry

The tissue microarray (TMA) was created using standard procedures that have been previously described [33,34]. From all patients included in this study, an experienced pathologist (FW) evaluated H&E sections of tumor tissues, and representative areas were marked. From these areas, core biopsies on the corresponding paraffin blocks were removed and transferred into the grid of a recipient block according to a predesigned array of about 60 specimens in each of the five TMA paraffin blocks. For immunohistochemistry, 4 μm sections of the TMA blocks were incubated with the indicated antibodies according to the mentioned protocols in the given dilutions (Table 2), followed by incubation with a horseradish peroxidase (HRP) conjugated secondary antibody and another incubation with 3,3′-diaminobenzidine (DAB) as substrate, which resulted in a brown-colored precipitate at the antigen site. An experienced clinical pathologist (FW) evaluated immunohistochemical staining according to localization and specificity (Table 3). For the determination of the staining intensity of ERRα and ERRγ, a score from 0 (negative) to 3 (strongly positive) was used. Since staining intensities for ERRβ were generally lower, a score from 0 to 2 was used. For steroid hormone receptors ERα, nuclear ERβ, and PR, the immunoreactivity score, according to Remmele et al., was used [35]. Expression of proliferation marker Ki-67 using antibody clone MIB-1 was assessed in the percentage of tumor cells with positive nuclear staining. Her2/neu expression was scored according to the DAKO score routinely used for breast cancer cases. EGFR was scored according to Spaulding et al. on a 4-tiered scale from 0 to 3 [36]. For p53 and polyclonal CEA, the “quick score” was used, where results are scored by multiplying the percentage of positive cells (*P*) by the intensity (*I*) according to the formula: *Q* = *P* × *I*; maximum = 300 [37]. CA-125 and ERβ were described as positive or negative, irrespective of staining intensity. Chemerin and CMKLR1 cellular staining intensity (non-specific nuclear staining was not considered) was scored on a 3-tiered scale from 1 (weak) to 3 (strong intensity) (Figure 1).

### 2.3. In Silico Analyses

To compare the expression of *RARRES2* and *CMKLR1* in normal ovary, OC, and OC metastases at the mRNA level, the TNMplot webtool (https://tnmplot.com/analysis/) was used to analyze gene chip data from GEO datasets, including 744 OC patients, 46 samples from the normal ovary and 44 OC metastases [38]. The statistical significance of the comparison was determined using the nonparametric Kruskal–Wallis test. To test the association of *RARRES2* and *CMKLR1* mRNA levels in OC patients with overall survival by means of the webtool KMplot (https://kmplot.com/analysis/index.php?p=service&cancer=ovar (accessed on 2 February 2023)), gene chip data from TCGA and 14 GEO datasets were analyzed. Both mRNA and survival data were available from 2021 OC patients. The following parameters were used for this analysis: splitting of the patients’ collective in a high and a low expression group was performed by choosing the “auto select best cutoff” option; all patient subgroups and treatment groups were included, and biased arrays were excluded. For *RARRES2*, the Affymetrix ID *209496_at* was indicated, and for *CMKLR1*, the Affy ID *210659_at* [39].

### 2.4. Statistical Analysis

Apart from multivariate survival analyses, statistical analysis was performed using GraphPad Prism 5^®^ (GraphPad Software, Inc., La Jolla, CA, USA). The non-parametric Kruskal–Wallis rank-sum test was used for testing differences in the expression among three or more groups. For pairwise comparison, the non-parametric Mann–Whitney U rank-sum test was used. Correlation analysis was performed using the Spearman correlation. Univariate survival analyses were performed using the Kaplan–Meier method. The chi-squared statistic of the log rank was used to investigate differences between survival curves. Hazard ratios were calculated using the Mantel–Haenszel method. A *p*-value below 0.05 was considered significant. Multivariate Cox regression survival analysis was performed using IBM^®^ SPSS^®^ Statistics 25 (SPSS^®^, IBM^®^ Corp., Armonk, NY, USA) using the Enter method.

## 3. Results

### 3.1. Intratumoral RARRES2 mRNA Levels in Ovarian Cancer and Metastasis Tissues Are Significantly Reduced When Compared to Normal Ovary

Given that a sufficient amount of normal ovarian tissues or metastatic tissues could not be obtained, it was decided to use the benefits of open-source gene chip expression data, and it was thereby possible to compare mRNA expression of *RARRES2* (coding for chemerin) and *CMKLR1* in 744 OC tissues, 46 samples from the normal ovary and 44 tissue samples of OC metastases. This analysis of open-source data using TNMplot webtool (https://tnmplot.com/analysis/) [38] accessed on 15. September 2022 revealed decreased *RARRES2* mRNA levels in the OC (Dunn test *p* = 0.0002) and the metastasis group (Dunn test *p* = 0.0646) compared to normal ovarian tissue, interpreted as an attempt for evasion from the immune response. Regarding *CMKLR1* mRNA levels, only the metastasis samples exhibited a reduced expression (Dunn test *p* < 0.0001) of this receptor (Figure 2).

### 3.2. Protein Levels of Chemerin and CMKLR1 in Ovarian Cancer Tissue

Both chemerin and CMKLR1 were shown to be widely detectable in OC tissues as assessed on the protein level by means of immunohistochemistry of tissue microarrays (TMAs). Positive staining of chemerin was found in all cases (32.7% with weak staining, 40.5% moderate, and 26.8% with strong staining). CMKLR1 was also detected in all tumors, among them 22.2% with weak staining, 38.0% with moderate, and 39.9% with strong staining. There was a strong correlation between chemerin and CMKLR1 levels in all tumors (rho = 0.5959, *p* < 0.0001), as well as the largest subgroup of serous OC (rho = 0.6285, *p* < 0.0001). No significant differences in protein levels of either chemerin or CMKLR1 between G2 and G3 graded tumors, different FIGO stages, or in patients with different nodal statuses were observed. Moreover, the invasion of lymph or blood vessels did not depend on the expression of either protein.

### 3.3. Protein Levels of Chemerin and CMKLR1 in Ovarian Cancer Tissue Subject to Levels of Ovarian Cancer Markers, Cancer-Related Proteins and Components of Estrogen Signaling Pathways

Subsequently, mean protein levels of chemerin and CMKLR1 in ovarian cancer subgroups were compared with high vs. low expression of the ovarian cancer markers, cancer-related proteins, and components of estrogen signaling pathways that were analyzed in this study.

First, results showed that mean levels of chemerin and CMKLR1 were elevated in ovarian cancers with higher cytoplasmic ERβ expression when compared to the lower expressing subgroup (*p* = 0.0143 and *p* = 0.0133, respectively) (Table 3). Mean protein levels of CMKLR1 were increased in ovarian cancer specimens with higher expression of the proliferation marker Ki67 (*p* = 0.0304). Protein levels of chemerin and CMKLR1 were elevated in the ERRα-high subgroup (*p* < 0.0001 and *p* < 0.0001, respectively). In ovarian cancers with higher expression of ERRβ, increased levels of chemerin and CMKRL1 (*p* = 0.0091 and *p* < 0.0001, respectively) were observed. CMKLR1 levels were found to be elevated in tumors with higher expression of ERRγ (*p* = 0.0031). Finally, the mean protein expression of chemerin was elevated in ovarian cancers with higher expression of CMKRL1 (*p* < 0.0001), and the mean protein levels of CMKRL1 was increased in ovarian cancer with higher expression of chemerin (*p* < 0.0001). No differences in chemerin and CMKLR1 expression levels could be observed between tumor subgroups with different levels of ERα, nuclear ERβ, PR, CEA, CA125, CA72-4, p53, Her2, or EGFR.

### 3.4. Correlation of Chemerin and CMKLR1 Protein Levels with Intratumoral Expression of Proteins Involved in Estrogen Signaling, Ovarian Cancer Markers, and Other Cancer-Related Genes

Since chemerin is known to affect ovarian steroidogenesis and was reported to correlate with steroid hormone receptors in breast cancer, correlations of both proteins with protein expression of PR, ERα, ERβ, PR, ERRα, β, and γ were examined first. Furthermore, intratumoral chemerin and CMKLR1 levels were tested for correlation with ovarian cancer markers CA125 (MUC16), polyclonal CEA (CEACAM1,3,4,6,7 and 8), and CA72-4 and with the cancer-related genes EGFR, HER2, Ki-67 and p53. By means of Spearman’s rank correlation analysis, a strong association of chemerin with progesterone receptor (PR) levels (Spearman’s rho = 0.7952, *p* < 0.0001) was observed. Chemerin and CMKLR1 were found to be moderately associated with intratumoral protein expression of ERβ, particularly in the largest serous subgroup, which was true both for nuclear (chemerin: rho = 0.2127, *p* = 0.0213; CMKLR1: rho = 0.2630, *p* = 0.0039) and cytoplasmic (chemerin: rho = 0.2731, *p* = 0.0029; CMKLR1: rho = 0.27, *p* = 0.003) ERβ expression. Notably, a considerable positive correlation between both chemerin and CMKLR1 with the estrogen-related receptors (ERR)s α, β, and γ was observed. Chemerin positively correlated with ERRα (rho = 0.384, *p* < 0.0001), ERRβ (rho = 0.3343, *p* < 0.0001), and ERRγ (rho = 0.383, *p* < 0.0001). CMKLR1 was associated with the expression of ERRα (rho = 0.5207, *p* < 0.0001), ERRβ (rho = 0.4239, *p* < 0.0001), and ERRγ (rho = 0.4198, *p* < 0.0001). Additionally, a weak positive association with cancer marker CEACAM5 (rho = 0.1594, *p* < 0.0498) was observed. Expression of the other proteins mentioned above was not significantly associated with either chemerin or CMKLR1 (Table 4).

### 3.5. Correlation of RARRES2 and CMKLR1 mRNA Levels with Expression of Genes Involved in Sex Steroid Hormone Metabolism and Signaling Assessed by In Silico Analysis

In silico analyses on the mRNA level (using gene chip data from 744 ovarian cancer patients accessed on the platform https://tnmplot.com) [38] on 15 September 2022 corroborated the positive correlation between chemerin (*RARRES2*) and *CMKLR1* that had been observed on the protein level (Spearman’s rho = 0.26, *p* < 0.0001). With regard to genes involved in estrogen signaling, this analysis also substantiated the positive correlation of *CMKLR1* with ERβ (*ESR2*) (rho = 0.33, *p* < 0.0001) and of *CMKLR1* with ERRα (*ESRRA*) (rho = 0.33, *p* < 0.0001), which was further corroborated using the GEPIA2 platform [40] analyzing datasets from 426 serous OC patients (*CMKLR1/ESR2* rho = 0.35 and *CMKLR1/ESRRA* rho = 0.31, both *p* < 0.0001). Using the same platform and data, a positive, albeit weaker correlation of *CMKLR1* with ERRβ (*ESRRB*) (rho = 0.2, *p* < 0.001) in serous OC, but not with ERRγ (*ESRRG*) was found. In contrast to the chemerin protein data from IHC, mRNA levels of the *RARRES2* gene in ovarian cancer were not correlated with *PGR, ESR2, ESRRA, ESRRB, ESRRG,* nor *CEACAM5* after analysis of both patient collectives on the mentioned platforms (*p* > 0.05 for all).

### 3.6. Survival Analyses

Association of chemerin and CMKLR1 in ovarian cancer tissue with overall and progression-free survival.

Analyzing the protein data assessed in this study by IHC of TMAs, when OC patients exhibiting different levels of intratumoral chemerin or CMKLR1 were compared with regard to OS by means of Kaplan–Meier analysis, no significant differences were found. Subsequently, the survival of patients with serous ovarian cancers was investigated. However, neither chemerin nor CMKLR1 levels did influence the OS of the patients in this cohort (Appendix A). The levels of these proteins also did not correlate with progression-free survival (PFS), neither when including all ovarian cancer cases nor when analyzing only serous ovarian cancers.

Since a weakness of this study is the relatively low number of OC samples, it was speculated that the association between chemerin and CMKLR1 expression with survival could be visible using a larger patient collective. Thus, the online tool kmplot.com providing microarray mRNA and OS data of 2021 OC patients from the Gene Expression Omnibus and The Cancer Genome Atlas [39] was used and accessed on 1 September 2022. This analysis revealed high mRNA levels of *RARRES2* in OC tissue to be significantly associated with a shorter OS (HR = 1.32, *p* = 5.8 × 10^−5^). In contrast, high mRNA expression of *CMKLR1* was associated with longer OS (HR = 0.8, *p* = 0.0002) (Figure 3).

## 4. Discussion

In this study, possible associations between the adipokine chemerin and its receptor CMKLR1 with other proteins involved in steroid hormone signaling were examined in OC tissues and in silico, as the role of these proteins in cancer is yet mostly unclear.It was found that in serous ovarian cancer, both chemerin and CMKLR1 protein positively correlated with ERβ protein expression and with levels of ERRα, β, and γ; additionally, chemerin protein expression was notably associated with that of PR. On the mRNA level, *CMKLR1*, not *RARRES2* mRNA, correlated with ERRβ and γ. These findings thus showed an association of chemerin/CMKLR1 with a nuclear estrogen receptor (ERβ), an important estrogen target gene (PR), and with modulators of estrogen signaling, which plays essential roles in OC.

Chemerin has been shown to modulate steroidogenesis, especially secretion of progesterone, in the porcine ovary in both stimulatory and inhibitory ways [41], and it has been proposed that chemerin via CMKLR1 plays a role in the development of polycystic ovary syndrome via inhibition of progesterone secretion [42]. Since progesterone is known to be of importance in OC development, the association between chemerin/CMKLR1 and PR was investigated. In our cohort of 208 patients, a strong correlation between chemerin staining intensity and PR protein expression could be shown. PR expression in OC was found to be associated with a more favorable prognosis [43], and further studies may confirm the role of chemerin herein.

It has long been demonstrated that estrogens, their different receptors (ERs), and related receptors (ERRs) are major players in the origin and development of OC in various ways, which led to an investigation of possible associations of chemerin and CMKLR1 with different ERs and ERRs, on which there are few data published to date. One study by Hoffmann et al. indicated an anti-proliferative effect of chemerin partly via ERs [44]. In our study, both chemerin and CMKLR1 levels in tumor tissues positively correlated with estrogen receptor β (ERβ), which could be confirmed on the mRNA level for CMKLR1 and ESR2 by in silico analysis. According to past publications, this could indicate a protective role of chemerin and CMKLR1 similar to ERβ [21,22,23,24].

Concerning ERRs, both chemerin and its receptor positively correlated with estrogen-related receptor α (ERRα), particularly in serous OC tissue, an association being also validated in silico on the mRNA level for *CMKLR1*. This is in line with a previous study [26], where ERRα was detected abundantly in OC tissues. Also, protein levels of chemerin and its receptor were associated with ERRβ and ERRγ, with a stronger correlation present in serous OC. As these two receptors are indicative of poorer survival [26], the exact mechanisms of chemerin interaction with ERRs and other modulatory factors are to be further elucidated since these findings are contradictory in their putative pro-tumoral effects to the association found with ERβ protein expression and ESR2 gene expression.

In silico analyses comparing mRNA expression of the *RARRES2* gene in normal ovary, OC, and OC metastases revealed a notable decrease of *RARRES2* expression in OC and in metastatic tissue, whereas *CMKLR1* RNA levels were considerably reduced in OC metastases only. Low expression of chemerin in tumor tissue is in accordance with findings from other cancer entities and was suggested to indicate a protective role of chemerin in cancer progression. Gao et al., however, described a higher expression of chemerin protein in OC compared to normal tissues. Intratumoral chemerin protein levels were not associated with the overall (OS) or progression-free survival (PFS) of OC patients. In line with our data, chemerin was found to be low-expressed in melanoma and liver cancer, but according to the Human Protein Atlas, it was not prognostic in these cancers [45]. Analysis of open-source mRNA and survival data from 2021 OC patients moreover identified a favorable effect of high *CMKLR1* and low *RARRES2* mRNA levels on patients’ survival. Taken together, the association of chemerin and CMKLR1 with ovarian cancer prognosis seems to be complex, and factors such as hormonal status or comorbidities such as adiposity, dyslipidemia, or hypertension must be considered.

The fact that an association of chemerin or CMKLR1 protein levels with OC survival was not observed, but instead, a significant correlation on the mRNA level of a larger patients´ collective might be explained by the different collective size. Furthermore, mRNA levels do not always correlate with the level of the coded protein. During phases such as cell proliferation or differentiation, post-transcriptional mechanisms may cause deviations from this association. The sampling of tissues for RNA and protein analysis is a further source of variations [46]. Chemerin is a secreted protein and may be taken up by cancer cells. Thus, there are different explanations for why mRNA and protein analysis of chemerin in OC did not always reveal concordant results. The first two arguments also apply to the further proteins analyzed in this study. For CMKLR1, it is important to note that only tumor cell expressed protein was quantified. At the mRNA levels, tumor cells, as well as further cells such as immune cells of the respective tissues, are included and contribute to variations of mRNA and protein data. Differences in protein level assessment of chemerin via immunohistochemistry and *RARRES2* gene expression on the mRNA level can be explained by the fact that chemerin is mainly produced by extratumoral tissues, e.g., adipocytes and hepatocytes [8]. Therefore, intratumoral protein levels measured by immunohistochemical staining are expectedly higher than mRNA levels when comparing normal and cancer tissues, and associations of intratumoral chemerin levels with OS and PFS are not mirrored by mRNA gene expression data.

Tumors including OC are able to escape the intrinsic anti-tumor activity of the immune system by means of so-called immune evasion strategies [47,48] and cancer immunoediting, often attributed to the interaction of tumor cells with tumor-infiltrating lymphocytes as well as immunomodulatory factors such as PD-L1, CTLA-4, and CXCR4 [49,50]. This might be a possible explanation for the missing effect of different intratumoral chemerin levels on OS or PFS, as well as the decrease of *RARRES2* on the mRNA level in the in silico analysis of OC, compared to normal ovarian tissue.

In this context, it might be of interest to investigate the composition of tumor-infiltrating lymphocytes and their interaction with chemerin via CMKLR1 in further studies.

Limitations of this study are the medium-sized cohort of OC patients and the lack of normal ovarian tissue in the immunohistochemical analysis, which has been compensated for in the additional in silico analyses on the mRNA level. As always in the case of adipokines and the like, it remains to be further determined how serum levels of chemerin must be taken into account, as serum chemerin levels were not available for our OC cohort.

## 5. Conclusions

Chemerin protein and its receptor CMKLR1 were demonstrated to be abundantly detectable by immunohistochemistry in ovarian cancer tissues and to positively correlate with intratumoral expression of PR, ERβ and ERRs, corroborating interaction with estrogen signaling pathways as previously suggested. Analysis of publicly available gene expression data demonstrated a significant downregulation of *RARRES2* mRNA expression in OC and metastatic tissue, whereas *CMKLR1* expression was found to be reduced in metastases only. Tumoral chemerin and CMKLR1 protein levels were not related to OS, but lower *RARRES2* and higher *CMKLR1* mRNA levels were associated with longer OS. Our data are able to encourage further studies examining the role of the interactions suggested in this study for the development and progression of ovarian cancer.

## Figures and Tables

**Figure 1 diagnostics-13-00944-f001:**
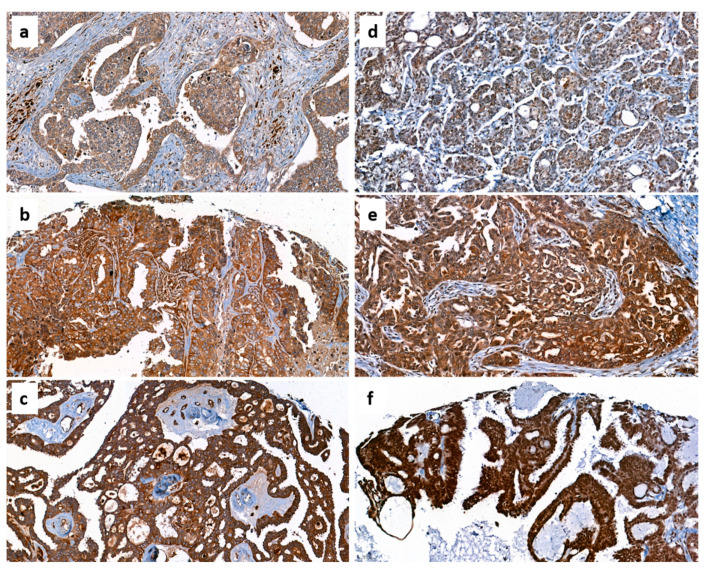
Examples of different chemerin and CMKLR1 immunohistochemical staining intensities in ovarian cancer tissues at 200× magnification. Chemerin expression ranging from 1 to 3 (**a**–**c**) and CMKLR expression ranging from 1 to 3 (**d**–**f**).

**Figure 2 diagnostics-13-00944-f002:**
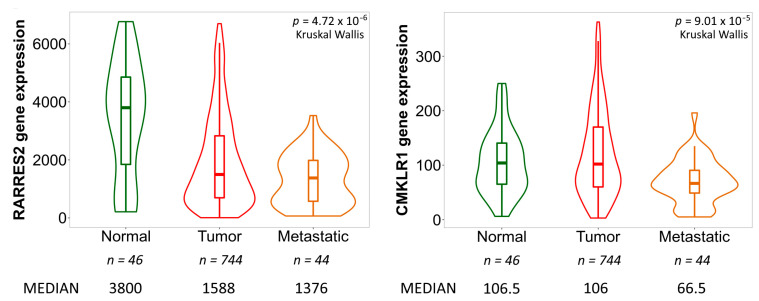
Comparison of *CMKLR1* and *RARRES2* mRNA expression in normal ovary, ovarian tumors, and ovarian cancer metastases by means of open-source gene chip data and the TNMplot webtool (https://tnmplot.com/analysis/) [38]. Expression levels are shown in TPM (transcripts per million). The number of patients in each group (*n*) and the median mRNA levels are indicated.

**Figure 3 diagnostics-13-00944-f003:**
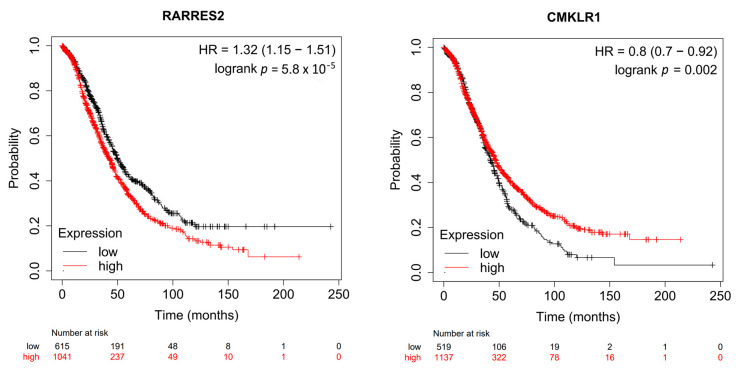
Association of *RARRES2* and *CMKLR1* mRNA expression with overall survival of ovarian cancer patients. Kaplan–Meier plots are shown based on microarray mRNA and overall survival data of 2021 ovarian cancer patients from the Gene Expression Omnibus and The Cancer Genome Atlas generated using the website kmplot.com [39].

**Table 1 diagnostics-13-00944-t001:** Characteristics of included patients and their tumors: Tissues from 208 Caucasian women with sporadic ovarian cancer were used in this study. The median age of the females was 64 (29–91) years. Serous ovarian cancers represent 64.90% of the tumors, and 58.65% were grade 3. Most of the cancers were diagnosed in FIGO (International Federation of Gynecologists and Obstetricians) stages III and IV (31.25% and 24.04%, respectively). During the median follow-up of 1180 days, 80 relapses and 62 deaths were documented. While median relapse-free survival was 1044 days, median overall survival (OS) was 1079 days.

Characteristics	Number of Patients	(%)
	208	100
**FIGO stage**		
FIGO I	22	10.58
FIGO II	8	3.85
FIGO III	65	31.25
FIGO IV	50	24.04
Unknown	63	30.29
**Histological subtype**		
Serous	135	64.90
Mucinous	6	2.88
Endometroid	10	4.81
Clear cell	3	1.44
Undifferentiated	54	25.96
**Histological grade**		
G2	53	25.48
G3	122	58.65
Unknown	33	15.87

**Table 2 diagnostics-13-00944-t002:** Antibodies used in this study (CC1: TRIS-EDTA borate buffer pH 8.0–8.5 at 95 °C P1: protease 1 (highest level) at 36 °C).

Marker/Protein	Antibody Clone	Pretreatment	Dilution	Pattern
**Chemerin (RARRES2)**	LS-B13333 (Biozol)	none	1:100	cytoplasmic (non-specific) nuclear
**CMKLR1**	LS-B12924 (Biozol)	none	1:100	membranous/cytoplasmic
**ERβ**	PPG5/10 (Novus Biologicals)	none	1:20	nuclear/cytoplasmic
**ERα**	6F11 (Novocastra)	CC1 64 min	1:35	nuclear
**CA-125**	OC125 (Cell Marque)	CC1 52 min	1:1	cytoplasmic/membranous
**CEA (polyclonal)**	A 0115 (Dako)	P1 8 min	1:500	cytoplasmic
**CA72.4**	B72.3 (Alexis Biochemicals)	CC1 36 min	1:50	cytoplasmic
**EGFR**	E30 (Dako)	P1 4 min	1:100	membranous
**p53**	sc-263 (Santa Cruz)	CC1 36 min	1:2000	nuclear
**Ki-67**	MIB-1/M7240 (Dako)	CC1 64 min	1:100	nuclear
**PR**	NCL-L-PGR-312 (Clone 16) (Novocastra)	CC1 64 min	1:50	nuclear
**Her2/neu**	A0485 (Dako)	CC1 36 min	1:250	membranous

**Table 3 diagnostics-13-00944-t003:** Correlation analysis of mean IHC staining intensity score of chemerin and CMKLR1 in ovarian cancer subject to high and low IHC score of the indicated proteins. The staining intensity score of chemerin and CMKLR1 was assessed in values between 0 (absent staining) and 3 (strong staining). Statistical significance was stated in the case of *p* < 0.05 and is highlighted by light grey color with a bold *p*-value; SD is shown in brackets. (ERβ (n) = nuclear, ERβ (cm) = cytoplasmic staining).

		Chemerin	CMKLR1			Chemerin	CMKLR1
**ER** **α**	low	1.902 (0.7822)	2.143 (0.7810)	**TP53**	low	1.910 (0.7926)	2.090 (0.7781)
	high	2.049 (0.7400)	2.195 (0.7490)		high	1.984 (0.7512)	2.238 (0.7559)
		*p* = 0.2878	*p =* 0.7393			*p =* 0.5472	*p =* 0.2451
**ER** **β (n)**	low	1.895 (0.7665)	2.119 (0.7736)	**HER2**	low	1.893 (0.7585)	2.139 (0.7640)
	high	2.429 (0.5345)	2.571(0.5345)		high	2.133 (0.8193)	2.200 (0.8052)
		*p =* 0.0683	*p =* 0.1372			*p =* 0.1355	*p =* 0.6702
**ER** **β (cm)**	low	1.838 (0.7423)	2.060(0.7576)	**EGFR**	low	1.905 (0.7739)	2.119 (0.7757)
	high	2.212 (0.7809)	2.424 (0.7513)		high	2.200 (0.6959)	2.450 (0.6863)
		***p =* 0.0143**	***p =* 0.0133**			*p =* 0.1069	*p =* 0.0752
**PR**	low	1.941 (0.7675)	2.169 (0.7655)	**ERRα**	low	1.671 (0.7082)	1.753 (0.6827)
	high	2.231 (0.7250)	2.308 (0.7511)		high	2.187 (0.7478)	2.533 (0.6438)
		*p =* 0.1925	*p =* 0.5397			***p <* 0.0001**	***p <* 0.0001**
**CEA**	low	1.908 (0.7889)	2.107 (0.7671)	**ERRβ**	low	1.524 (0.6016)	1.524 (0.6016)
	high	2.143 (0.6547)	2.429 (0.7464)		high	2.000 (0.7776)	2.246 (0.7477)
		*p =* 0.1839	*p =* 0.0684			***p =* 0.0091**	***p <* 0.0001**
**CA125**	low	1.950 (0.8256)	2.000 (0.8584)	**ERRγ**	low	1.632 (0.7609)	1.632 (0.8307)
	high	1.940 (0.7663)	2.180 (0.7571)		high	1.970 (0.7611)	2.220 (0.7343)
		*p =* 0.9701	*p =* 0.3678			*p =* 0.0691	***p =* 0.0031**
**CA72.4**	low	1.955 (0.7756)	2.188 (0.7656)	**CMKLR1**	low	1.314 (0.5827)	
	high	1.925 (0.7642)	2.100 (0.7779)		high	2.127 (0.7226)	
		*p =* 0.8374	*p =* 0.5349			***p <* 0.0001**	
**Ki-67**	low	1.936 (0.7488)	2.083 (0.7592)	**Chemerin**	low		1.560 (0.6440)
	high	2.031 (0.8224)	2.406 (0.7560)		high		2.447 (0.6527)
		*p =* 0.5490	***p =* 0.0304**				***p <* 0.0001**

**Table 4 diagnostics-13-00944-t004:** Correlations of intratumoral protein levels of chemerin and CMKLR1 assessed by IHC of tissue microarrays (TMAs) with a total of 208 OC tissue samples with protein expression of estrogen receptor α (ERα), estrogen receptor β (ERβ), progesterone receptor (PR), proliferation marker Ki67, erb-b2 receptor tyrosine kinase 2 (Her2/neu), epidermal growth factor (EGFR), p53, carcinoembryonic antigen (CEA), cancer antigen 72-4 (CA72-4), and the estrogen-related receptors α, β, and γ (ERRα, ERRβ, and ERRγ) are shown using Spearman´s rank correlation analysis for all OCs and the subgroup of serous OC. In the case of ERβ, different results for nuclear (n) and cytoplasmic/membranous (cm) staining are stated. *p*-values below 0.05 were considered significant. n.s. = no significant correlation.

	Ovarian Cancer		Serous Ovarian Cancer	
	Chemerin	CMKLR1	Chemerin	CMKLR1
**ERα**	n.s.	n.s.	n.s.	n.s.
**PR**	*p* < 0.0001 rho = 0.7952	n.s.	*p* < 0.0001 rho = 0.8175	n.s.
**Ki67 (MKI67)**	n.s.	n.s.	n.s.	n.s.
**CA-125 (MUC16)**	n.s.	n.s.	n.s.	n.s.
**Her2**	n.s.	n.s.	n.s.	n.s.
**EGFR**	n.s.	n.s.	n.s.	n.s.
**p53**	n.s.	n.s.	n.s.	n.s.
**CEA (CEACAM1,** **3,4,6,7 and 8)**	*p* = 0.0498 rho = 0.1549	n.s.	*p* = 0.0428 rho = 0.1868	n.s.
**CA72-4**	n.s.	n.s.	n.s.	n.s.
**ERβ (n)**	n.s.	*p* = 0.0009 rho = 0.2641	*p* = 0.0213 rho = 0.2127	*p* = 0.0039 rho = 0.2630
**ERβ (cm)**	*p* = 0.0137 rho = 0.2009	*p* = 0.007 rho = 0.216	*p* = 0.0029 rho = 0.2731	*p* = 0.003 rho = 0.2700
**ERRα**	*p* < 0.0001 rho = 0.384	*p* < 0.0001 rho = 0.5207	*p* < 0.0001 rho = 0.3989	*p* < 0.0001 rho = 0.4709
**ERRβ**	*p* < 0.0001 rho = 0.3343	*p* < 0.0001 rho = 0.4239	*p* = 0.0007 rho = 0.3082	*p* < 0.0001 rho = 0.3665
**ERRγ**	*p* < 0.0001 rho = 0.3830	*p* < 0.0001 rho = 0.4198	*p* < 0.0001 rho = 0.4534	*p* < 0.0001 rho = 0.4869

## Data Availability

Data can be obtained from the authors upon reasonable request.

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
