# Peer review of "Chemerin and Chemokine-like Receptor 1 Expression in Ovarian Cancer Associates with Proteins Involved in Estrogen Signaling"

_diagnostics, 2023, doi:10.3390/diagnostics13050944_

Round 1
Reviewer 1 Report
The manuscript discusses a novel topic - protein levels of chemerin and CMKLR1 assessed by immunohistochemistry of tissue microarrays and their association with patients´ survival and with an expression of ovarian cancer markers was tested.
However, before publishing the text must be improved. Please find below my comments:
1. The title is too long and difficult to comprehend. It has to be reorganized and rephrased.
2. the absract clearly represents the study. The writing style couls be improved. For example origina sentence in lines 15-17 "To further approach the role of this adipokine in ovarian cancer (OC), we examined intratumoral protein levels of chemerin and its receptor chemokine-like receptor 1 (CMKLR1) by immunohistochemistry analyzing tissue microarrays with tumor samples from 208 OC patients." Suggested: "To further approach the role of this adipokine in ovarian cancer (OC), intratumoral protein levels of chemerin and its receptor chemokine-like receptor 1 (CMKLR1) were examined by immunohistochemistry analyzing tissue microarrays with tumor samples from 208 OC patients."
3. The introduction part is well-written and provides the study rationale clearly and in full.
4. The material and methods part should include the following parts - study design, study subjects, study tools, and study variables, ethical considerations.
5. The results part is interesting and supported by high-quality tables and figures/pictures.
6. In general, the discussion part is interesting, however, requires improvement to meet the style appropriate for research manuscripts. The first paragraph of the discussion should provide a potential reader with the study rationale in brief. Please find below the suggestions on how the discussion part could be restructured:
Discussion
1.1 Rationale of the study (why it was done)
1.1.1 Main findings of the study
1.1.2 What makes our study unique
1.1.3 What it adds to what we already know
1.2 Subject of the discussion
Comparison of your results with neighboring countries, with countries of the same
development levels (income), with developed high-income countries). Agreement and disagreement with the studies compared
1.4 Sum up the study, study strengths, and limitations
1.5 Clinical implication
General comments
1. The text should be written from a third-person point of view and written in a classical academic writing style appropriate for a research paper.
2. Check the text for grammar and spelling mistakes as there are a lot of them.
Please provide a thorough point-by-point response to all comments of the reviewers and the Editor. Responses like “Done”, “ Revised”, etc. will not be accepted.
Reviewer 2 Report
The authors studied the correlation of chemerin and CMKLR1 protein expression levels with other tumor markers and estrogen signaling protein levels in ovarian cancer patients. The results are novel and can be interesting for scientific community. However, the following concerns should be considered.
Major concerns
1. For the in-silico results described (with TNMplot and kmplot), the detailed description should be provided in the Materials and Methods section. KMplot includes many parameters, and all of them should be listed. Also, it would be helpful to shortly describe the type of the data in these databases.
2. For lines 282-283, if the Kaplan–Meier analysis was performed, the results should be provided in Figure (e.g., in Supplementary Materials).
3. For each correlation analysis, scatter plot in Supplementary Materials would be helpful to understand the spread of values and their correlation.
Minor concerns
Line 109 – median age or average age?
Line 127 – HRP should be defined.
Table 3 – the caption should be rephrased. There is a correlation analysis between the variables, therefore the type of analysis should be defined in the caption
Round 2
Reviewer 2 Report
The manuscript can be accepted in the present form